# The Effect of Prenatal Exposure to Climate Anomaly on Adulthood Cognitive Function and Job Reputation

**DOI:** 10.3390/ijerph19052523

**Published:** 2022-02-22

**Authors:** Hong Tang, Qian Di

**Affiliations:** 1Vanke School of Public Health, Tsinghua University, Beijing 100084, China; hongtang@tsinghua.edu.cn; 2Institute for Healthy China, Tsinghua University, Beijing 100084, China

**Keywords:** El Niño/Southern Oscillation (ENSO), sea surface temperature anomalies, cognitive function, job reputation, prenatal exposure, salary loss

## Abstract

Background: The long-term effect of abnormal climate on cognitive function and socioeconomic status remains elusive. We explored the association between prenatal exposure to climate anomaly and adulthood cognitive function and job reputation. Methods: We obtained repeated cognitive and job reputation measurements from 17,105 subjects for the years 2010, 2014, and 2018, and ascertained their birth date and other covariates. We used sea surface temperature (SST) anomalies in the Southern Pacific Ocean as the indicator for global climate anomaly in the main analyses. We calculated its averaged values for different gestational periods and analyzed its possible nonlinear associations with adulthood cognitive function and job reputation. We also calculated associated economic loss due to prenatal exposure to abnormal climate. Results: We found an inverted U-shaped curve between climate anomaly and adulthood cognition. During the entire pregnancy, for SST anomalies increasing/decreasing 1 °C from 0 °C, newborn individuals will have adulthood cognition (measured by math test) changed by −2.09% (95% confidence interval (CI): −2.31%, −1.88%) and −3.98% (95% CI: −4.32%, −3.65%), respectively. We observed a similar inverted U-shaped pattern for cognitive function measured by word test and job reputation. Such an association is likely to be mediated by regional meteorological conditions, not local ones. Subgroup analyses identified females and people from less-developed regions as even more vulnerable to prenatal abnormal climate, finding an interactive effect with other social factors. The economic loss was assessed as the salary reduction due to declined cognition among all newborn individuals in China. For SST anomalies increasing/decreasing by 1 °C from 0 °C, individuals born each year in China would earn 0.33 (95% CI: 0.40, 0.25) and 1.09 (95% CI: 1.23, 0.94) billion U.S. dollars equivalent less in their annual salary at adulthood because of lowered cognitive function, respectively. Conclusion: Prenatal exposure to abnormal global climate patterns can result in declined adulthood cognitive function, lowered job reputation, and subsequent economic loss.

## 1. Introduction

Human health and earth climate are closely correlated, and abnormal climate could induce impaired human health, which is the core idea of planetary health [1]. Featured by warming and unfavored temperature, climate change and its associated health effect have attracted much research interests [2]. Unfavored temperature, including heatwave and cold waves, leads to various adverse health outcomes, including reduced labor capacity [3], suboptimal cognitive function [4], altered DNA methylation level [5], elevated incidence of cardiovascular disease [3], and eventually increased mortality [6], among others. Most related studies have focused on the daily and short-term effect of local temperature [6], although there is some recent evidence on the health effect of temperature variation [7] as well as in the longer terms [8].

In addition to unfavored temperature, abnormal climate also includes unfavored precipitation pattern, humidity, and other indicators, making infectious disease and famine more likely, which will indirectly affect human health [1]. Thus, the health impact of climate cannot be attributable to a single meteorological variable (i.e., temperature); El Niño/Southern Oscillation (ENSO), a periodic variation closely related to El Niño and La Niña, captures some type of overall global climatic pattern and serves as an ideal indicator for assessing the complex indirect health effect of climate. Following this idea, some researchers analyzed the health effect of ENSO, including hospital admission for diarrheal disease [9], childhood malnutrition [10], hemorrhagic fever [11], malaria transmission [12], and other vector-borne diseases [13], but they still focused on the short-term effect of ENSO. Overall, existing studies primarily focused on the short-term health effect (e.g., daily or weekly level) of local weather conditions [1], especially thermal stress, unfavored temperature, or ENSO. However, a few studies took a holistic point of view and explored the long-term effect (e.g., decade level) of climate at a broad geographic scale (e.g., global scale) and analyzed the associated indirect social costs.

Therefore, in this study, we attempted to explore the association between prenatal/early life climate conditions and adulthood cognition and job reputation. We used ENSO indexes as indicators of climate pattern at global scale. Cognitive function was measured at multiple times at adulthood. We used a mixed-effect model to estimate the long-term health effect of prenatal and early life climate exposure on adulthood cognitive function and job reputation. To identify vulnerable population, we conducted subgroup analyses in terms of socioeconomic status, regional development level, sex, age, and lifestyle. To explore possible influencing pathways, we conducted mediation analysis and estimated the mediation effect by weather conditions. Lastly, we estimated economic cost associated with aberrant prenatal and early life climate by calculating salary loss at adulthood due to impaired adulthood cognitive function. We believe this study can remind people of the long-term health effects of global climate pattern, as well as its indirect social and economic cost, and moreover can encourage actions on climate mitigation.

## 2. Materials and Methods

### 2.1. Study Design and Data Source

In this longitudinal study, we obtained individual-level information from the China Family Panel Studies (CFPS) with 17,105. CFPS is a large-scale social follow-up survey, with national representative samples of Chinese families and individuals to capture changes in China’s society, economy, population, education, and health. This study used deidentified data and was approved by CFPS office.

### 2.2. Cognitive Ability Tests, Job Reputation Scores, and Covariates

CFPS used cognitive tests to quantify cognitive function. The cognitive ability tests include math test (24 standardized mathematical questions) and word test (34 word-recognition questions). Questions were sorted by difficulty level and easier questions were asked first. The cognitive test was finished when respondents make three wrong answers consecutively. The final test score was the number of correct answers a respondent makes. The full math and word test scores were 24 and 34, respectively. Respondents’ major job was surveyed; job reputation was quantified on the basis of the Standard International Occupational Prestige Scale (SIOPS), which reflects the general evaluation and overall recognition of a job position. The job reputation score from [14] ranges from 0 to 100, with higher score corresponding to higher prestige. Other covariates were also extracted from the CFPS dataset, including gender, age, smoking status, income level, physical exercise frequency, and the wave of follow-up. 

### 2.3. Climate Anomaly and Meteorological Data

We used ENSO to define climate anomaly. There are several ENSO indexes, and we used sea surface temperature (SST) anomalies in the Southern Pacific Ocean 5° N–5° S, 120–170° W (i.e., Niño 3.4), as the indicator of climate anomaly in the main analyses. This is an indicator commonly used to define El Niño or La Niña events: SST anomalies ≥0.5 °C indicates an El Niño event, while ≤−0.5 °C indicates a La Niña event. For each subject, we calculated SST anomalies for the entire pregnancy, each trimester, and each three-month period within one year after birth, on the basis of the birth date. We also used other ENSO indexes to measure climate anomaly as sensitivity analyses (Appendix A). This is in consideration of the fact that SST anomalies in the tropical Pacific requires some time to affect the global climate system and influence China. Thus, we used ENSO indexes lagged by 2 months in the main analysis, consistent with a previous study [15]. Air temperature (°C) and precipitation were obtained from European Center for Medium-Range Weather Forecasts, ERA5 (Fifth generation ECMWF atmospheric reanalysis) dataset [16]. We calculated monthly averaged temperature/precipitation for each province and county and matched it to each individual on the basis of the city of residence. 

### 2.4. Statistical Analyses

To account for longitudinal measurements of cognitive function and job reputation, we used a mixed-effect model with random effect on the individual. In the follow-up mixed-effect model analyses, dependent variables were math test score, word test score, and job reputation score; the independent variable was prenatal or early-life exposure to climate anomaly, defined as SST anomalies at different time periods. This model was adjusted for fixed effects of age, gender, income, smoking status, and physical exercise frequency, and random effect for individual. To account for any nonlinear relationship, we put B-spline with 3 degrees of freedom as the smooth function on SST anomalies in the mixed-effect model. To estimate economic loss associated with prenatal exposure to climate anomaly, we used a methodology previously used to monetize the economic loss of prenatal exposure to lead [17]. Briefly speaking, first, we used mixed-effect model with spline to assess the nonlinear relationship between climate anomaly and cognitive function. Second, we also assessed the association between cognitive function and income. Thus, we established indirect quantitative association between prenatal climate anomaly and adulthood income and used income change to evaluate economic loss associated with prenatal exposure to climate anomaly. We also performed mediation analysis to investigate the mechanism by which a climate anomaly affects adulthood cognitive function. We considered temperature and precipitation at county- and province-levels as potential mediators, since temperature and precipitation are the two most commonly used meteorological variables in epidemiological studies. All analyses were performed in R software (R version 4.1.0; R Foundation for Statistical Computing). 

## 3. Results

A nationally representative sample of 17,105 subjects born between 1950 and 1994 from China were included in the analysis (Table 1, Appendix A). We found a long-term association between prenatal and early life exposure to global climate anomaly and adulthood math test score. More interestingly, this relationship demonstrated an inverted U-shaped pattern. For the entire pregnancy, SST anomalies increasing to 1 °C from 0 °C (i.e., prone to El Niño events) were associated with −2.09% (95% CI: −2.31%, −1.88%) change in math test score, and decreasing to −1 °C (i.e., prone to La Niña events) was associated with −3.98% (95% CI: −4.32%, −3.65%) change in math test score (Figure 1). By examining the association in three month periods, we identified second and third trimesters of pregnancy, 7–9 months after birth, and 10–12 months after birth to be the critical periods (Appendix A). Compared with math test score, the impact of climate anomaly on word test score also followed the inverted U-shaped pattern, despite of the impact at lesser magnitude (Figure 1 and Appendix A). Thus, we focused on the math test score as an indicator for cognitive function in the following analyses; results related to the word test are included in the Appendix A.

Apart from adulthood cognitive function, we also found a similar inverted U-shaped pattern between adulthood job reputation and pregnancy/early life exposure to global climate anomaly (Figure 1 and Appendix A). People prenatally or postnatally exposed to abnormal global climate pattern were more likely take low-level jobs in adulthood. This is partially because one’s cognitive function can impact his/her competitiveness in the job market and job attainment [18], and our dataset also indicated a positive correlation between cognitive function and career development (Appendix A). If abnormal climate results in suboptimal adulthood cognitive function, it will also subsequently affect career development and job reputation. Collectively, we found nonlinear relationships between global climate anomaly during pregnancy and early life with adulthood cognitive function and job reputation, a social status indicator.

Global climate anomaly is defined as SST anomalies in the Southern Pacific Ocean, at least 10,000 km from our study area. To investigate the underlying mechanism, we analyzed the mediation effect by local temperature and precipitation, since several existing studies found that weather conditions during pregnancy can affect offspring health [19]. We found significant indirect effects mediated by province-level temperature (*p* = 0.02) and precipitation (*p* = 0.03), explaining 7.97% (*p* = 0.05) and 18.64% (*p* = 0.07) of the association between climate anomaly and adulthood cognition (Figure 2b,d). In contrast, the mediation effects by county-level temperature and precipitation were not significant (Figure 2a,c), indicating provincial meteorological conditions to be better mediators. It suggests that the long-term health effect of global climate anomaly is more likely to be mediated by regional weather conditions, rather than local weather conditions. 

The cognitive impact of climate anomaly may not be evenly distributed in the population. To identify populations vulnerable to climate anomaly, we conducted subgroup analyses. We found females and people from less-developed regions to be more vulnerable to abnormal climate (Figure 3A,D). The inverted U-shaped pattern was more pronounced for females, while such a pattern was not significant for males (Figure 3A), and the impact of climate anomaly on cognition was attenuated in developed regions (Figure 3D). Interestingly, the impact of climate anomaly during pregnancy did not diminish with aging; in contrast, the aging process seemed to amplify the cognitive impact of prenatal exposure to climate anomaly (Figure 3B). Furthermore, subgroup analyses indicated that climate anomaly interacted with some acquired factors and jointly affected adulthood cognition; smoking severely impairs one’s cognitive function [20], and thus the prenatal impact of climate anomaly was less significant in this subgroup. Lower income level also amplified the prenatally impact of climate anomaly (Figure 3E). In addition, compared with regions weakly connected to ENSO, the impact of climate anomaly during pregnancy on adulthood cognitive ability were more remarkable in regions teleconnected to ENSO (Figure 3F).

To further explore the robustness of the association, we (1) explored the same model specification with different time lag (Appendix A), (2) used the Southern Oscillation Index as the indicator for climate anomaly (Appendix A), and (3) used outgoing longwave radiation as the indicator for climate anomaly (Appendix A). These sensitivity analyses all yielded a similar inverted U-shaped pattern, demonstrating the robustness of our results. 

Prenatal and early life abnormal climate affects adulthood cognitive function, limits career development, and ultimately imposes a large social cost. We monetized the economic cost of abnormal climate following a previous study on lead exposure and IQ [17]. First, we assessed the nonlinear association between prenatal exposure to climate anomaly and adulthood cognitive function. Second, we assessed the association between cognitive function and income. Third, we calculated annual income loss associated with prenatal exposure to climate anomaly by person. Fourth, we calculate economic loss as the aggregated income loss among people who were prenatally exposed to climate anomalies (Appendix A). Individuals born during a period with SST anomalies increasing/decreasing 1 °C from 0 °C would end up with −20.75 (95% CI: −25.36, −16.14) and −69.16 (95% CI: −78.38, −59.94) dollar equivalent change in annual salary in adulthood, respectively. Since we used a national representative sample of China, and we can apply these numbers to all newborn individuals in China every year (e.g., 15.74 million in 2018), we found SST anomalies increasing/decreasing 1 °C from 0 °C would end up with 0.33 (95% CI: 0.40, 0.25) and 1.09 (95%CI: 1.23, 0.94) billion U.S. dollars in equivalent loss (Figure 4, Appendix A). Using the word test as an indicator for cognitive function yielded economic loss at a similar magnitude (Appendix A).

## 4. Discussion

In this study, we analyzed the association of adulthood cognitive function and job reputation with abnormal climate conditions during pregnancy and in early life. We found a long-term association between climate anomaly and suboptimal adulthood cognitive function and job reputation. This association was better explained by regional weather conditions, rather than local ones. Females and people from less-developed regions were more vulnerable to prenatal exposure to climate anomaly. Prenatal climate anomaly can interact with age, income, and lifestyle, and jointly affects cognitive function and job reputation. We also conducted sensitivity analyses using other climate indicators and achieved similar results. Overall, prenatal and early life climate imposes long-term effect on human cognitive function, resulting in economic loss. For SST anomalies, increasing/decreasing 1 °C from 0 °C results in −0.33 (−0.40, −0.25)/−1.09 (−1.23, −0.94) billion dollars equivalent salary reduction as economic loss. We hope to use this study to remind people of the long-term health effect of climate patterns and encourage people to take actions to mitigate climate change. 

Our study found an association between prenatal climate anomaly and adulthood cognitive function and job reputation. Although this association has not been reported before, several existing studies have already implied possible mechanisms: on one hand, prenatal malnutrition promotes metabolic dysfunction and phenotypes of obesity and diabetes via epigenetic pathways: genes related to birth weight and serum LDL cholesterol are differentially methylated [21] and eventually affect cognition [22]. On the other hand, climate pattern can influence agricultural production, including wheat, rice, maize, soybean, and other staples [23], and multiple agricultural yields. Thus, prenatal and early life climate pattern can affect offspring health by food availability, food diversity, and other nutritional factors. Furthermore, parental income also influences offspring cognitive function, and such an effect persists into adulthood [24]. Meanwhile, economic growth is correlated with climate pattern [25], and aberrant climate patterns negatively affect economic growth and have profound social and economic impacts [26]. Thus, climate pattern during pregnancy can affect local economic growth, family income, and eventually influence offspring cognitive function decades later. Therefore, existing knowledge supports an association between climate anomaly and adulthood cognition. Moreover, some previous studies have already found a short-term health impact of climate anomaly, indicated by ENSO, including hospital admission for diarrheal diseases [9] and growth stunting [27]. On this basis, we further extended this ENSO–health connection to a greater time span at the decade level from prenatal exposure to adulthood cognitive function, with a greater geographic coverage. 

We also identified critical periods and further proved the biological plausibility of climate–cognition association. A Dutch famine study revealed that those exposed to malnutrition during pregnancy have higher prevalence of obesity, altered lipid metabolism, elevated incidence of cardiovascular disease, and reduced cognitive function in later life, and the impact of malnutrition was attenuated after birth, suggesting a critical period of malnutrition affecting normal development of metabolism system and cognition function [28]. These studies imply gestation to be the critical window of cognitive development. Unfavored environment at this critical period affects normal development of fetus profoundly, and such an adverse effect would even persist throughout their life. Our study also revealed a similar pattern: pregnancy exposure to climate anomaly was associated with declined adulthood cognition. In a slightly different way, we also found early life exposure to climate anomaly, shortly after birth, attenuated adulthood cognition, although at a lesser magnitude. In summary, both the Dutch famine study and our study indicate gestational period as the critical period of cognitive development. In addition to that, climate anomalies can affect heatwave frequency, and prenatal exposure to heatwave can increase risks of miscarriage and stillbirth [29,30]. Climate anomalies also affect infectious disease spreading, e.g., malaria, whose infection during pregnancy has adverse effects on both mother and fetus and may also affect fetus in the long run. Thus, the inverted U-shaped relationship between climate anomaly and adulthood cognitive function is biological plausible.

The impact of prenatal climate pattern is not equally distributed, and people born in less developed provinces are more vulnerable to unfavored prenatal climatic pattern. This result is achieved after controlling for income level, suggesting that regional economic development independently interacts with climatic pattern and affects offspring cognitive function. Less developed provinces are already lagging behind the national average in terms of school education, parental education, nutrition, and many other aspects. Now, economic underdevelopment interacts with abnormal climate and places these vulnerable people in an even more disadvantaged position. Although this study was limited only within China, we believe this result also applies to other countries as well. However, our study also gives people hope: developed provinces, on the contrary, can use advantages in these aspects to compensate the impact of unfavored climate, and similarly, policymakers can leverage economic development to reduce or even eliminate the impact of abnormal climate.

Placing the social implication aside, this study also has some academic implications and demonstrates the importance of a holistic point of view while analyzing the health impact of climate. On the basis of our mediation analysis, weather conditions only explain a small portion of the association between prenatal climate anomaly and adulthood cognition, about 7.97% is mediated by temperature. In addition, weather conditions from a large geographic coverage (e.g., provincial temperature) are a better mediator. This means that current epidemiological paradigm analyzing the health impact of *local* weather conditions only captures a small portion of potential health impact of climate, and the overall health impact of climate is underexplored and remains poorly understood. Thus, it is important to adopt a holistic point of view: we should (1) take climatic patterns from broader areas into consideration, (2) focus on the long-term health effects, and (3) consider social effects as well. The current epidemiological paradigm focuses on the causal relationship between each exposure–outcome pair, but as demonstrated by our study, the impact of climate on human health can undergo multiple pathways that impose several methodological challenges to traditional epidemiological studies. New research paradigms are needed to tackle this challenge. 

This study is among the first studies to address the association between prenatal climate pattern and offspring health, and thus inevitably bears several limitations. First, this study only focused on cognitive function, without considering other health outcomes; thus, our understanding about the health burden of abnormal climate is limited to cognitive function. However, according to the Dutch famine study, people exposed to prenatal famine had reduced adulthood cognition and metabolic dysfunction. Thus, we should assume that impaired cognitive function may be suggestive of other adverse health effect as well. Second, this study was limited only to China, but we believe this study is generalizable to other countries as well: China covers areas with various geographic, weather, and economic conditions, and thus captures at least some portion of the complex climate–health association. Results obtained from China are somehow representative and should be generalized to other countries. Our study also has several strengths. We are among the first to identify the association between prenatal climate anomaly and offspring cognition, which implies that macro-climate pattern could have a long-term effect on human health. This not only deepens our understanding on the interaction between human health and climatic system but also suggests the importance of a holistic point of view while analyzing climate–health issues. Moreover, this study also reminds people of the long-term and previously underexplored health impact of abnormal climate and suggests the importance of immediate actions to mitigate climate change.

## 5. Conclusions

Prenatal and early life abnormal climate affects adulthood cognitive function, limits career development, and finally imposes a large social cost. We found an inversed U-shaped curve between their prenatal and early life exposure to abnormal climate and adulthood cognitive function and their job reputation. Therefore, policymakers and the general public must consider such a threat and should take action right now to mitigate climate change.

## Figures and Tables

**Figure 1 ijerph-19-02523-f001:**
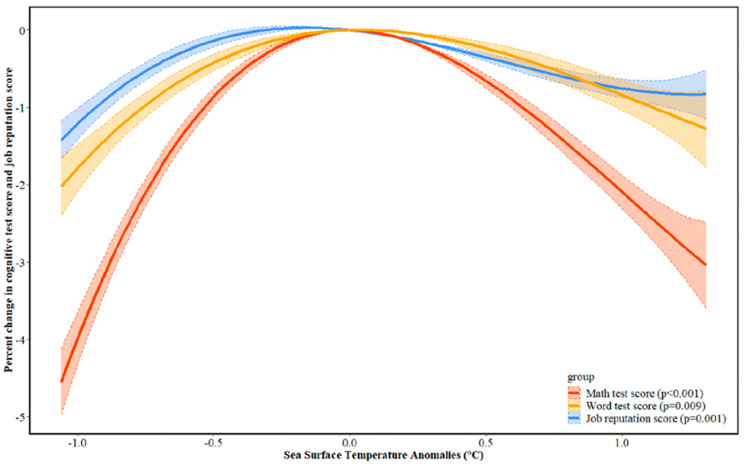
Dose–response relationship between prenatal exposure to climate anomaly and adulthood cognitive function and job reputation. Note: We used a mixed-effect model with random effect on individual to calculate dose–response relationship. In the mixed-effect model analyses, dependent variables were math test score, word test score, and job reputation score; independent variable was 2 month lagged SST anomalies in entire pregnancy. This model was adjusted for fixed effect of age, gender, log-transformed income, smoking status, physical exercise frequency, and a random effect for individual. To account for any nonlinear relationship, we used B-spline with 3 degrees of freedom as the smooth function on SST anomalies in the mixed-effect model. We converted changes in cognitive test score/job reputation score into percent change by dividing the mean value. Percent changes in cognitive test score and job reputation score were centered at SST anomalies = 0 °C. Dose–response curves outside 5th and 95th percentiles of SST anomalies were trimmed. To assess the sensitivity of our results to the choice of degree of freedom, we conducted sensitivity analysis with df = 4 and 5, respectively (Appendix A).

**Figure 2 ijerph-19-02523-f002:**
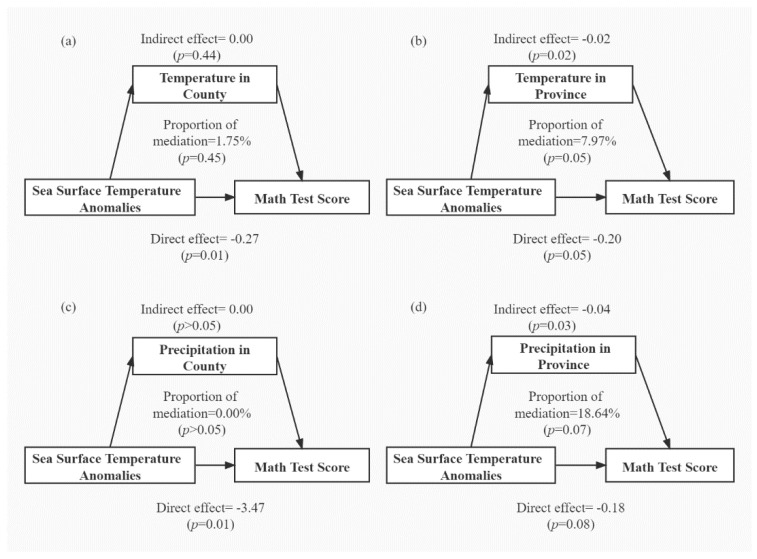
Mediation effect by regional and local meteorological conditions (**a**) county-level temperature, (**b**) province-level temperature, (**c**) county-level precipitation, (**d**) province-level precipitation. Note: The mediation effect was estimated using the “mediation” and “mgcv” package in R software, with statistical significance assessed by quasi-Bayesian Monte Carlo approach with 1000 times of simulation. In the outcome model, we used generalized additive model with math test score as the dependent variable and SST anomalies in entire pregnancy as the independent variable and adjusted for fixed effects of mediator to be tested, age, gender, log-transformed income, smoking status, physical exercise frequency, and the wave of follow-up. We used B-spline as the smooth function on mediator variable and independent variable to account for any nonlinearity. In the mediator model, we used generalized additive model with mediator variable as the dependent variable and SST anomalies in entire pregnancy as the independent variable. This model was also adjusted for fixed effect of age, gender, log-transformed income, smoking status, physical exercise frequency, and the wave of follow-up. We also used B-spline as the smooth function on independent variable in the mediator model.

**Figure 3 ijerph-19-02523-f003:**
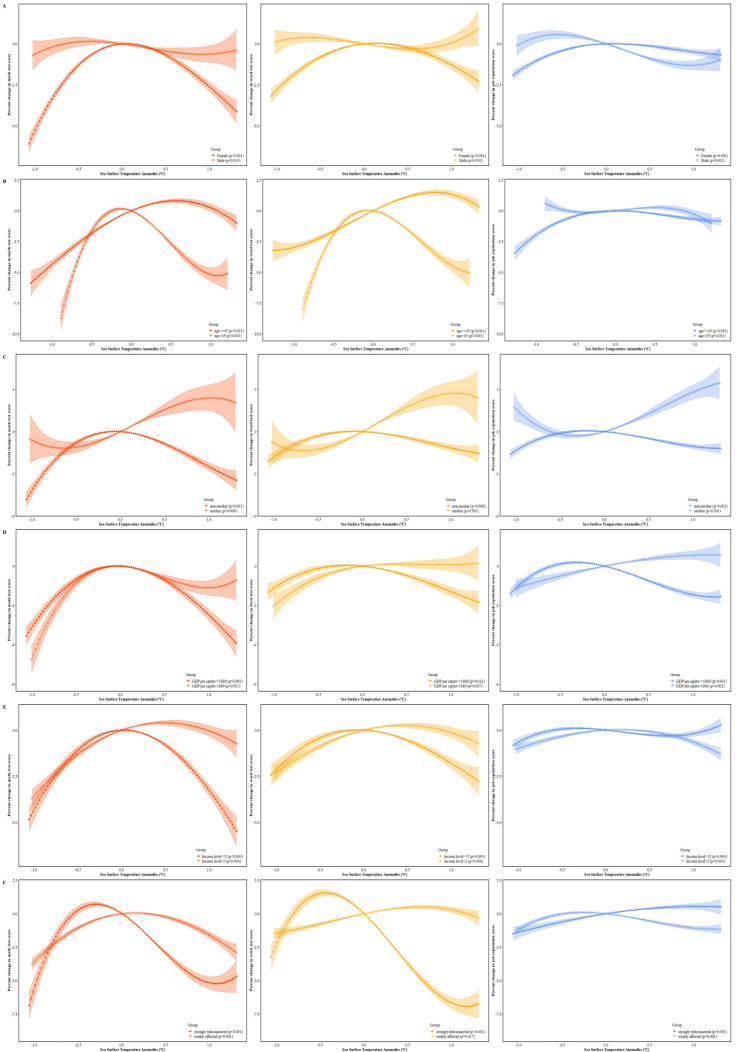
Dose–response relationship between adulthood cognitive function, job reputation, and prenatal climate anomaly by subgroup: (**A**) by gender, (**B**) by age, (**C**) by smoking status, (**D**) by GDP per capita, (**E**) by income level, and (**F**) by region associated with ENSO. Note: To obtain dose–response curves by subgroups, we repeated the same model as in Figure 1 for each subgroup. We converted change in cognitive test score and job reputation into relative scale by dividing the mean values. Percent changes in cognitive test score and job reputation score were centered at SST anomalies = 0 °C. Dose–response curves outside 5th–95th percentile of SST anomalies were trimmed. The method for ENSO-teleconnection partition was found in Appendix A.

**Figure 4 ijerph-19-02523-f004:**
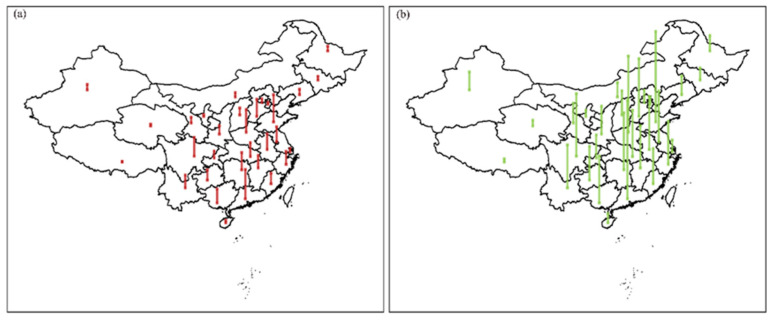
Economic loss due to prenatal exposure to climate anomaly in different regions of China: (**a**) SST anomalies increasing 1 °C from 0 °C; (**b**) SST anomalies decreasing 1 °C from 0 °C. Note: we used salary loss to monetize economic loss due to prenatal exposure to SST anomalies. For Figure 4, we only plotted economic loss mediated by declined math test score. Economic loss mediated by word test score was visualized in Appendix A.

**Table 1 ijerph-19-02523-t001:** Descriptive analysis of demographic information and exposure profile.

	N/Observations	Math Test (Mean ± SD)	Word Test (Mean ± SD)	Job Reputation (Mean ± SD)
Demography information in the entire population	17,105/19,204	10.63 ± 6.57	18.48 ± 10.54	40.10 ± 11.65
By gender				
Female	12,702/14,298	9.91 ± 6.60	17.46 ± 10.86	39.69 ± 11.33
Male	4405/4906	12.73 ± 6.00	21.45 ± 8.90	41.09 ± 12.30
By age				
Age ≤ 45	10,921/12,007	11.91 ± 6.31	20.88 ± 9.66	40.90 ± 11.63
Age > 45	6804/7197	8.48 ± 6.43	14.48 ± 10.71	38.54 ± 11.53
By smoking status				
Smokers	1301/1302	11.77 ± 5.61	20.36 ± 8.98	39.92 ± 11.28
Non-smokers	15,877/17,902	10.54 ± 6.62	18.34 ± 10.63	40.12 ± 11.68
By income level ^a^				
Income level ≤ 2	9077/9461	10.04 ± 6.41	17.65 ± 10.57	38.74 ± 10.31
Income level > 2	7926/8642	11.45 ± 6.54	19.71 ± 10.21	41.14 ± 12.55
By GDP per capita (USD) ^b^				
GDP per capita ≤ 1880	11,034/12,279	10.14 ± 6.62	17.54 ± 10.76	40.19 ± 11.08
GDP per capita > 1880	6075/6925	11.48 ± 6.39	20.13 ± 9.91	39.97 ± 12.52
By region associated with ENSO ^c^				
Weakly affected	13,934/15,679	10.92 ± 6.53	19.20 ± 10.34	40.24 ± 11.88
Strongly teleconnected	3174/3525	9.33 ± 6.58	15.28 ± 10.80	39.55 ± 10.58

Note: ^a^ Individual income was categorized from 1 to 5, with 1 standing for the lowest income level and 5 for the highest. ^b^ We used GDP per capita for the year 2014 in the subgroup analysis, and 6.38 as the exchange rate from Chinese Renminbi to US dollar. We used the same GDP per capita data and exchange rate in the follow-up analysis. ^c^ The method for ENSO–teleconnection partition is specified in Appendix A.

## Data Availability

Input data can be found from CFPS office upon request.

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
