# Peer review of "The Effect of Prenatal Exposure to Climate Anomaly on Adulthood Cognitive Function and Job Reputation"

_ijerph, 2022, doi:10.3390/ijerph19052523_

Round 1
Reviewer 1 Report
General remarks
The article is interesting and important. Conclusion is missing
Specific comments
- 12, please underline that you used SST as an ENSO index, “as the indicator of climate anomaly in the main analyses”.
- 46: According to the definition of climate (https://en.wikipedia.org/wiki/Climate), the temperature is one of the meteorological variables. The other are humidity, atmospheric pressure, wind, and precipitation. Such an assumption is precisely the starting point of your article. But here, it is not expressed in a clear-cut manner. Please rephrase the sentence.
- 63-64: prenatal/early-life climate conditions (please add)
- 131: why did you use only temperature and precipitations as mediators? Please explain
- 364: Conclusions are missing
Reviewer 2 Report
Paper is presenting a nationally representative sample of 17105 subjects born between 1950 to 1994 from China in order to examine the association between prenatal exposure to climate anomaly and adulthood cognitive function and job reputation.
The task is extremely ambitious as no much research was done in this area so far.
The major finding is an association between prenatal and early-life exposures to global climate anomaly and adulthood math test score, which demonstrates an inverted U-shaped pattern. A similar inverted U-shaped pattern was found between adulthood job reputation and pregnancy/early-life exposure to global climate anomaly. The Authors interpretation was that people prenatally or postnatally exposed to abnormal global climate pattern were more likely take low-level jobs at adulthood.
In order to identify vulnerable population to climate anomaly, The Authors conducted subgroup analyses. It was found that female and people from less-developed regions to be more vulnerable to abnormal climate The impact of climate anomaly on cognition was attenuated in developed regions .
Furthermore, subgroup analyses indicated that climate anomaly interacted with some acquired factors and jointly affects adulthood cognition. Lower income level amplified the prenatally impact of climate anomaly . Smoking was found to severely impair one’s cognitive function thus the prenatal impact of climate anomaly was less significant in this subgroup
Major remarks:
- The Authors are trying to link the exposure to climate anomaly with adulthood cognitive function and job reputation. The assumption is that climate anomalies negatively influence the maternal diet and as a result the fetus's development. However, no convenience rationale is based. The only analogy provided is the Dutch famine case.
- Several covariates, like smoking and low social-economic status, were incorporated in the analyses. However, no attempt was made to investigate the role of both household and outdoor pollution. Any attempt to approach this problem will be valuable.
Round 2
Reviewer 1 Report
Thank you for your changes.
I want to share just one comment which is crucial to me. In the conclusion section, it should be essential to mention that policymakers must consider such a threat and initiate initiatives that should start now. Because what you are describing has been caused in the past, hence we cannot wait for long term actions
Author Response
Thank you for your comments. We have revised this sentence. Please see Page 10 Line 373-Line 375.
